# Elevation and latitude drives structure and tree species composition in Andean forests: Results from a large-scale plot network

Agustina Malizia[1]☯*, Cecilia Blundo[1]☯, Julieta Carilla[1]☯, Oriana Osinaga Acosta[1]☯, Francisco Cuesta[2,3], Alvaro Duque[4], Nikolay Aguirre[5], Zhofre Aguirre[6], Michele Ataroff[7], Selene Baez[8], Marco Calderón-Loor[2,9], Leslie Cayola[10,11], Luis Cayuela[12], Sergio Ceballos[1], Hugo Cedillo[13], William Farfán Ríos[14], Kenneth J. Feeley[15], Alfredo Fernando Fuentes[10,11], Luis E. Gámez Álvarez[16], Ricardo Grau[1], Juergen Homeier[17,18], Oswaldo Jadan[13], Luis Daniel Llambi[8], María Isabel Loza Rivera[10,19,20], Manuel J. Macía[21,22], Yadvinder Malhi[23], Lucio Malizia[24], Manuel Peralvo[3], Esteban Pinto[3], Sebastián Tello[20], Miles Silman[25], Kenneth R. Young[26]

1 Instituto de Ecología Regional (IER), Universidad Nacional de Tucumán (UNT) - Consejo Nacional de Investigaciones Científicas y Técnicas (CONICET), Tucumán, Argentina, 2 Grupo de Investigación en Biodiversidad, Medio Ambiente y Salud (BIOMAS), Universidad de Las Américas (UDLA), Quito, Ecuador, 3 Consorcio para el Desarrollo Sostenible de la Ecorregión Andina (CONDESAN), Quito, Ecuador, 4 Departamento de Ciencias Forestales, Facultad de Ciencias Agrarias, Universidad Nacional de Colombia, Sede Medellín, Medellín, Colombia, 5 Centro de Investigaciones Tropicales del Ambiente y la Biodiversidad, Universidad Nacional de Loja, Loja, Ecuador, 6 Herbario Reinaldo Espinoza, Universidad Nacional de Loja, Loja, Ecuador, 7 Instituto de Ciencias Ambientales y Ecológicas (ICAE), Facultad de Ciencias, Universidad de Los Andes, Mérida, Venezuela, 8 Escuela Politécnica Nacional, Quito, Ecuador, 9 Centre for Integrative Ecology, School of Life and Environmental Sciences, Deakin University, Victoria, Australia, 10 Herbario Nacional de Bolivia (LPB), La Paz, Bolivia, 11 Missouri Botanical Garden, St, Louis, MO, United States of America, 12 Área de Biodiversidad y Conservación, Universidad Rey Juan Carlos, Madrid, España, 13 Facultad de Ciencias Agropecuarias, Universidad de Cuenca, Cuenca, Ecuador, 14 Herbario Vargas (CUZ), Universidad Nacional de San Antonio Abad del Cusco, Cusco, Perú, 15 Department of Biology, University of Miami, Florida, United States of America, 16 Laboratorio de Dendrología, Facultad de Ciencias Forestales y Ambientales, Universidad de Los Andes, Mérida, Venezuela, 17 Plant Ecology and Ecosystems Research, University of Gottingen, Gottingen, Germany, 18 Centre of Biodiversity and Sustainable Land Use (CBL), University of Gottingen, Gottingen, Germany, 19 Department of Biology, University of Missouri, Columbia, MO, United States of America, 20 Center for Conservation and Sustainable Development, Missouri Botanical Garden, St, Louis, MO, United States of America, 21 Departamento de Biología, Área de Botánica, Universidad Autónoma de Madrid, Madrid, España, 22 Centro de Investigación en Biodiversidad y Cambio Global (CIBC-UAM), Universidad Autónoma de Madrid, Madrid, España, 23 Environmental Change Institute, School of Geography and the Environment, University of Oxford, Oxford, England, United Kingdom, 24 Facultad de Ciencias Agrarias, Universidad Nacional de Jujuy, Jujuy, Argentina, 25 Center for Energy, Environment and Sustainability, Winston-Salem, North Carolina, United States of America, 26 Department of Geography and the Environment, University of Austin Texas, Texas, United States of America

☯ These authors contributed equally to this work.
* agustinamalizia@yahoo.com

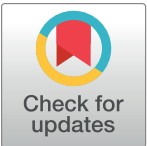

**Data Availability Statement:** All relevant data are within the manuscript and its Supporting Information files.

## Abstract

Our knowledge about the structure and function of Andean forests at regional scales remains limited. Current initiatives to study forests over continental or global scales still have important geographical gaps, particularly in regions such as the tropical and subtropical Andes. In this study, we assessed patterns of structure and tree species diversity along ~ 4000 km of latitude and ~ 4000 m of elevation range in Andean forests. We used the Andean Forest Network (Red de Bosques Andinos, https://redbosques.condesan.org/)

**Funding:** FC, MP received funding from Andean Forest Program implemented by the Consortium for the Sustainable Development of the Andean Ecoregion (CONDESAN) and Helvetas Swiss Intercooperation, and funded by the Swiss Agency for Development and Cooperation (SDC). Additional funding came from the EcoAndes Project conducted by CONDESAN and United Nations Environment Programme (UN Environment), funded by the Global Environment Facility (GEF; Cooperation Agreement No. 4750). AM, CB, LRM received funding for PICT-O 2014-0059 Project, FONCYT, Argentina. FC received funding from Universidad de las Américas for covering the publication fee and make this research open access.

**Competing interests:** The authors have declared that no competing interests exist.

database which, at present, includes 491 forest plots (totaling 156.3 ha, ranging from 0.01 to 6 ha) representing a total of 86,964 identified tree stems $\geq$ 10 cm diameter at breast height belonging to 2341 identified species, 584 genera and 133 botanical families. Tree stem density and basal area increases with elevation while species richness decreases. Stem density and species richness both decrease with latitude. Subtropical forests have distinct tree species composition compared to those in the tropical region. In addition, floristic similarity of subtropical plots is between 13 to 16% while similarity between tropical forest plots is between 3% to 9%. Overall, plots ~ 0.5-ha or larger may be preferred for describing patterns at regional scales in order to avoid plot size effects. We highlight the need to promote collaboration and capacity building among researchers in the Andean region (i.e., South-South cooperation) in order to generate and synthesize information at regional scale.

## Introduction

The Tropical Andes have been identified as one of the most important hotspots of global biodiversity [1, 2]. Indeed, this region is one of the most diverse terrestrial hotspots on Earth [1]. There are approximately 45,000 identified vascular plant species occurring in the Andes, 20,000 of which are endemic to the region [1]. There are also likely to be many more species (up to 35%) that have not yet been described [3]. Moreover around 60 million people depend directly or indirectly on ecosystem services provided by Andean forests, such as water and the regulation of regional climates [4, 5, 6, 7, 8].

Andean forests face a high risk of degradation as a result of climate change [9, 10] and land use change [11, 12] due to human population growth [7] and migration [12, 13, 14, 15], and a combination of these factors [15]. Changes in Andean forest cover includes both forest regrowth mostly frequent above 2000 m asl and deforestation that often concentrates below 1000 m asl [12]. Some Andean countries have already lost ~50–60% (~550.500 km$^2$ total) of their cloud forests [16] due to historical and ongoing land use changes, linked to the expansion of the agricultural frontier, increase in extension of cattle ranching areas for pastures, and other economic activities such as mining, road constructions among others [17, 18, 19]. Furthermore, as a result of warming, some forests and grasslands are experiencing an apparent shift in plant species composition (i.e. increasing relative abundances of species from lower elevations) [10, 20]. Moreover, model projections for biomes [21] and vascular plant species [22] predict upward shifts and the loss of optimal conditions in their lower and mid ranges. These changes may compromise the persistence of Andean forest ecosystems and may reduce the provision of benefits for human populations.

Recently, Mathez-Stiefel et al. [23] developed a research agenda for Andean forests landscapes where they highlighted critical knowledge gaps on the ecology of Andean mountain ecosystems, and the effect of socio-environmental changes over forest functions and dynamics. This includes the need to understand general patterns of tree diversity along elevational and latitudinal gradients, as well as forest structure and forest dynamics across environmental gradients at continental scales throughout integrated and collaborative research initiatives [23].

Tree or stem density and basal area may increase with elevation in extra Andean tropical montane forests [24, 25, 26, 27, 28]. However, in Andean forests, our understanding of structural parameters is limited to a handful of locations that have reported differing trends [29, 30, 31]. For example, in the tropical section of the Andes, basal area may increase with altitude until mid-elevations [32], and then decrease [30] or remain stable [29]. In subtropical Andes,

basal area tended to increase with elevation [31]. Baez et al. [33] made an effort to describe regional patterns of Andean forests structure and dynamics along elevation and latitudinal gradients. These authors found that in tropical mountain forests, basal area increased at lower elevation but did not change along the elevation gradient. They also found that in subtropical Andes, changes in basal area appear to be influenced by land use history together with environmental variation [33].

Plant species richness and diversity are thought to decline away from the equator and towards higher latitudes and elevations [34, 35]. However, elevational patterns of species richness have been studied much more intensively than latitudinal ones [34]. For example, a hump-shaped diversity pattern has been reported for various plant groups along elevational gradients in the Andes, including vascular epiphytes in Bolivia [36], non-vascular bryophytes in Colombia [37], and ferns in Ecuador and Bolivia [35]. Tree species richness (as well as tree genera and family richness) also shows the hump-shaped pattern, reaching a maximum at intermediate elevations (i.e. 1000 m asl) in subtropical Andes of Argentina [38], while decreasing above 1800 m asl in tropical Andes of Ecuador [30]. Still, regional-scale studies of tree community structure and composition along broader environmental gradients in the Andes are lacking.

This study is one of the first attempts to describe and characterize regional patterns of forest structure and diversity in Andean forest communities across ~ 4000 km of latitude and ~ 4000 m of elevation gradient. In this sense, this study incorporates a higher dataset than Baez et al. [33] (i.e. 8 times more) as well as diversity patterns. Specifically, for structural patterns we assessed: stem density and basal area, and for diversity patterns we assessed: species richness, tree species composition and floristic similarity along these elevation and latitudinal gradients. To accomplish this, we used the Andean Forest Network, AFN (Red de Bosques Andinos, https://redbosques.condesan.org/) database that provides data on forests throughout the tropical and subtropical Andes. Our main hypothesis is that the climatic gradient generated by elevation and latitude model the structural and diversity patterns of Andean forests, with plot size playing a role in explaining these trends. Specifically, we hypothesis that: (1) Tree communities located near the equator will have higher species richness, higher stem density and higher basal area than those located further away from the equator; and (2) Tree communities from lower elevations have a higher species richness, lower stem density and lower basal area than those located at higher elevations.

## Materials and methods

### Andes origin and vegetation

The Andes Mountains originated through a major uplift of the Central Andes during the Paleogene (65 to 34 Ma) and subsequent plates collisions intensified mountain building in the Northern Andes (23 Ma) [39]. However, the Andes reached their modern elevation during the late mid Miocene (~12 Ma) and early Pliocene (~4.5 Ma) [39, 40]. The formation of the Andes mountains had an enormous effect on regional climate regulation and was crucial for the evolution of landscapes and ecosystems in South America, including current biodiversity patterns of Amazon ecosystems [39]. For example, the uplift of the cordillera changed the Amazonian landscape by re-configuring the patterns of drainage and creating a vast influx of sediments into the basin [39]. Recent Pleistocene glaciations (110,000 to 10,000 b.p.) were also important in determining Andean forest biodiversity and species distributions [41]. For example, by acting as a refuge for biodiversity and by allowing immigration of holarctic species (e.g. *Alnus*), not only due to changes in temperature and $CO_2$ worldwide, but also due to variations in precipitation over the Amazon which had direct influence in Andean forests [42].

The Andean mountain forest ecosystem zonation is expressed in changes of forest architecture (i.e. decreases in tree stature and stem diameter, trends in stem deformation, hard, thick and smaller leaves; Fig 1) and tree community composition [16, 29, 30, 32, 33, 43] as elevation increases. The tropical and subtropical slopes vary in floristic composition and forest structure, from premontane forests (also called sub-andean forests) at lower elevations (from 500/800 to 700/1500 m asl) toward lower montane forests (1500/2700 m asl), upper montane forests (2700/2800 to 3300 m asl), and finally the upper treeline forests at higher elevations (3300/ 3500 -exceptionally 4000 m asl (Fig 1) [44]. Elevation ranges and vegetation zonation vary according to latitude, environmental humidity and topography [44]. For example, in the Andes of Colombia and Ecuador, moist inter-Andean valley slopes present similar gradations in vegetation zones. Further, vegetation zonation is related to changes in the frequency and persistence of cloud formation, which affects solar radiation, air temperature, and water regimes. Cloud formation varies from low cloud cover in premontane forest towards more persistent cloudiness in upper montane forest [16, 45, 46].

## Andean Forest Network

**Background and standardized protocol of measurement.** The AFN was created in 2012, being the first Network at this regional scale, and brings together scientists and decision makers interested in research, management and conservation of Andean forests. The main objective of the AFN is to generate knowledge about the ecology of the Andean forests throughout the collaborative work of its members, exchange, systematization and synthesis of information, development of research protocols, strengthening of research capabilities, and the articulation with decision making processes in the region. Specifically, the Andean Forest Network aims to: 1) promote the establishment and maintenance of long-term research sites through the installation and monitoring of permanent vegetation plots; 2) enhance the ability to understand the structure, composition, dynamics and functioning of Andean forests in a global context of environmental change; 3) promote collaboration and capacity building among researchers and technicians in the Andean region (i.e. South-South cooperation). For general information about the Network see https://redbosques.condesan.org/.

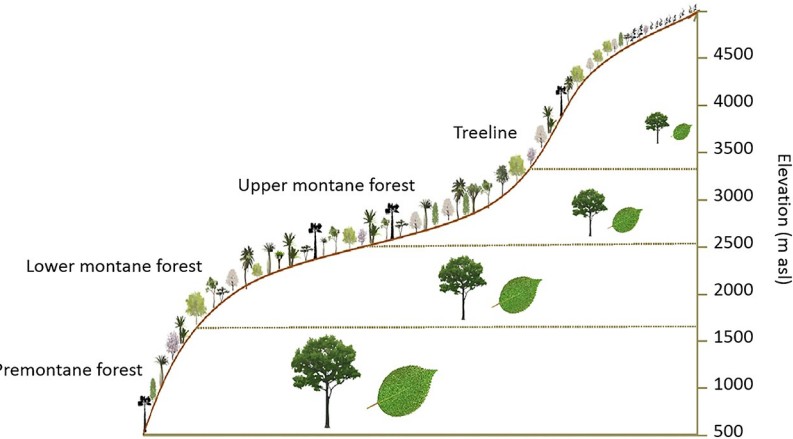

**Fig 1. Andean mountain forest zonation.** Scheme of mountain forest zonation in the elevational gradient expressed in forest architecture along tropical and subtropical Andes. Adapted from [46].

Members of the Andean Forest Network have detailed forest tree data (i.e. species identity, diameter, spatial coordinates) derived from research plots (both permanent and non-permanent). The Network prioritizes collaborative efforts amongst network members, encouraging them to share their data under confidence and transparent concepts, with previous consent and approval of the principal researchers and/or institutions [i.e. 10, 33, 47, 48, 49].

An important contribution of the Andean Forest Network has been the development of protocols for the establishment of permanent monitoring plots of plant diversity adapted to the conditions prevalent in the Andean forests [47, 48] which provide methodological approaches to study forest structure and dynamics of tree communities. Osinaga Acosta et al. [47], also includes an approach to consider and measure lianas, palms, ferns and herbs. For trees (with diameter ≥10 cm at breast height, DBH), the protocol encourages the establishment of 1-ha permanent forest plots subdivided in 20 × 20 m quadrants, where all individuals are identified to the lowest taxonomic level, marked with a numbered tags, and mapped in a x-y coordinate system to 1-m resolution. Botanical samples are collected and incorporated into local institutional herbaria. Plots are periodically recensused during which time mortality and recruitment events are recorded and the DBH of marked trees is remeasured. Recensus intervals vary among research objectives and available resources, although a five-year interval is suggested in order to maintain consistent monitoring over time [47].

**Forest plot distribution, general information and climate data.** At present, the Andean Forest Network comprises 491 forest plots spanning a latitudinal range from tropical Venezuela (8.63˚N) to subtropical Argentina (26.77˚S), a longitudinal range from the Pacific to the Atlantic versants (-80.14 to -63.84˚ W), and an elevation range from 41 to 3980 (mean ± SD = 1901 ± 903) m asl (Fig 2). Forest plots setup were based on the combination of several factors: (1) accessibility, (2) sustainable over time to ensure long term monitoring for some plots, and (3) homogeneity in terms of forest type, topography, and disturbances in order to be representative of particular forest types. The majority of the plots are located in mature forests but the AFN also includes several older secondary forests plots, i.e. > 30 years old [50] as a result of human activities and the associated land use changes [19, 51, 52, 53]. Censuses of plots were conducted between 2002 and 2017 (mean year of census = 2010.5±0.26) with the exception of one plot censused in 1988. The 491 plots range in size from 0.01 to 6 ha (mean plot size ± SD = 0.32 ± 0.47) totalizing 156.3 ha of which 68 plots are located in Argentina (62.7 ha; mean plot size ± SD = 0.92 ha ± 0.69); 27 in Bolivia (27 ha; mean plot size = 1 ha); 16 in Colombia (16 ha; mean plot size = 1 ha); 331 in Ecuador (34.4 ha; mean plot size ± SD = 0.10 ha ± 0.16); 46 in Peru (15.2 ha; mean plot size ± SD = 0.33 ha ± 0.37), and 3 in Venezuela (1.08 ha; mean plot size = 0.36 ha). Of the total of plots, 73% (n = 356) are permanent plots, of which 62% (n = 221) have been remeasured, in the majority of cases two to three times. Exceptionally, 22 plots (15 ha) in Argentina have been measured up to 6 times (i.e. 25 years of data) being the longest known subset for long-term studies of subtropical Andean forests (Fig 2, Table 1, S1 Table). Considering DBH cut off points, 47% of forest plots (n = 229) measured trees ≥10 cm, 37% (n = 184) measured trees ≥5 cm, 12% of forest plots (n = 60) measured trees ≥2.5 cm, and 4% (n = 18) measured trees ≥1cm (S1 Table). All relevant data upon which all the presented results in this manuscript are based on is included in the Supporting Information files; number of stems, basal area (both extrapolated to 1-ha), and species richness for the 491 plots in S1 Table while the abundance of species per country in S1 Appendix.

We extracted the following climatic data for each of the 491 plots from the CHELSA (Climatologies at high resolution for the earth's land surface areas) dataset at 30 arc sec (~1 km) resolution (period 1970–2013) [54]: estimated ranges in total annual rainfall varied from 608 to 5059 mm yr$^{-1}$ while estimated ranges in mean annual temperature varied from 5.9 to 27˚C (Table 1, S1 Table). Estimated mean temperature for the three coldest months varied between

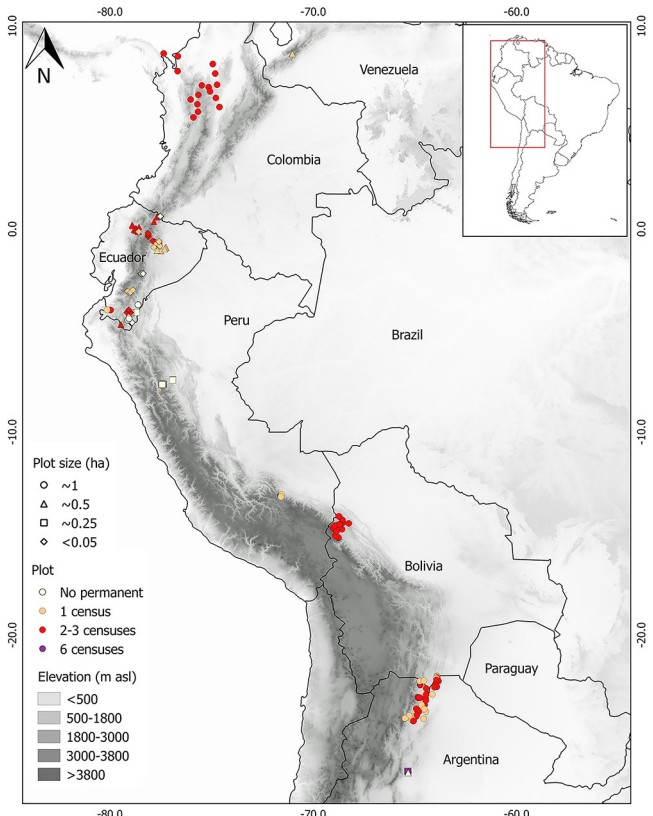

**Fig 2. Distribution of Andean Forest Network plots.** Forest plots (permanent and non-permanent) of the Andean Forest Network located along the Andean forests of Argentina, Bolivia, Peru, Ecuador, Colombia and Venezuela (n = 491). Different symbols and colors represent plot size and number of censuses, respectively. Plot size categories refer to: <0.05 (n = 235; 10 plots of 0.01, 205 plots of 0.04, 20 plots of 0.05), ~0.25-ha (n = 87; 20 plots of 0.08-ha, 60 of 0.1-ha, and 7 plots of 0.24-ha), ~0.5-ha (n = 57; 49 plots of 0.36-ha, 6 of 0.40-ha, and 1 plot of 0.48-ha), and ~1-ha (n = 112; 109 plots of 1-ha, 2 of 0.96-ha, and 1 plot of 6-ha).

-1 and 24°C, while the estimated mean temperature for the three warmest months varied from 11 to 32°C. Predicted seasonal thermal amplitude (mean temperature from the warmest quarter - mean temperature from the coldest quarter) varied from 1°C in Ecuador, Colombia and Venezuela to 10°C in Argentina, showing intermediates values in Peru (2°C) and Bolivia (4°C), depicting a clear latitudinal trend in thermal seasonality such that the thermal amplitude increases with distance from the equator.

## Data analyses

To analyze patterns in stem density, basal area and species richness, we considered those plots $\geq$ 0.1-ha (n = 236) where stems $\geq$10 cm DBH were measured. Due to high variability and dispersion when extrapolating basal area and stem density at a larger spatial scale we discarded plots of 0.01-ha, 0.04-ha, 0.05-ha and 0.08-ha (total n = 255). For those plots $\geq$ 0.1-ha, we estimated stem density and basal area at 1-ha for comparison while species richness was evaluated on a per plot basis. For calculations of stem density and basal area we used all stems $\geq$10 cm DBH (n = 97,054) while for species richness we used stems $\geq$10 cm DBH identified to species level (n = 86,964). We performed generalized linear models (GLM) [55] and used elevation, absolute latitude, and plot size as explanatory variables. We included the quadratic

**Table 1. Andean Forest Network metadata synthesis.**

| Countrya | N°Plots(area) | N°Permplots | Location S | | Location W | | Elevation (m asl) | | Area (ha) | | Rainfall mm | | Temperature °C | | Censuses # | | yrs | | MinDBH cm | | Stems* plot | | Basalarea* m² plot | | Spprichness* # | |
|---|---|---|---|---|---|---|---|---|---|---|---|---|---|---|---|---|---|---|---|---|---|---|---|---|---|---|
| | | | min | max | min | max | min | max | Min | max | min | max | min | max | min | max | min | max | min | max | min | max | min | max | min | max |
| AR | 68 (62.7) | 63 | -22.0 | -26.8 | -63.8 | -65.5 | 396 | 2304 | 0.16 | 6.00 | 608 | 1693 | 12 | 22 | 1 | 6 | 1992 | 2017 | 5 | 10 | 78 | 1834 | 5.0 | 189.4 | 3 | 52 |
| BO | 27 (27) | 27 | -14.2 | -22.2 | -64.6 | -69.0 | 662 | 3324 | 1.00 | -- | 644 | 1792 | 9 | 23 | 1 | 2 | 2003 | 2016 | 10 | -- | 430 | 1099 | 17.1 | 37.1 | 16 | 113 |
| PE | 46 (15.2) | 16 | -4.7 | -13.2 | -71.6 | -79.5 | 745 | 3450 | 0.10 | 1.00 | 903 | 2007 | 10 | 25 | 1 | 2 | 1998 | 2017 | 2.5 | 10 | 33 | 1262 | 2.3 | 45.8 | 13 | 107 |
| EC | 331 (34.3) | 231 | 0.7 | -4.6 | -77.2 | -80.1 | 360 | 3980 | 0.01 | 1.00 | 790 | 5059 | 6 | 25 | 1 | 3 | 2000 | 2017 | 1 | 10 | 1 | 660 | 0.02 | 44.7 | 1 | 118 |
| CO | 16 (16) | 16 | 5.5 | 8.7 | -74.6 | -77.4 | 41 | 2928 | 1.00 | -- | 1854 | 4354 | 12 | 27 | 2 | 3 | 2006 | 2014 | 1 | -- | 372 | 1245 | 18 | 40.6 | 31 | 120 |
| VE | 3 (1.1) | 3 | 8.6 | -- | -71.0 | -71.0 | 2300 | 2700 | 0.36 | -- | 1016 | 1043 | 13 | 14 | 1 | -- | 2017 | -- | 5 | -- | 323 | 332 | 14.7 | 16.1 | 37 | 39 |

aCountry codes are AR, Argentina; BO, Bolivia; PE, Peru; EC, Ecuador; CO, Colombia; and VE, Venezuela. Location (S, W), elevation (m asl), area (ha), total annual rainfall (mm), mean annual temperature (°C), censuses (#, years), minimum DBH (cm), number of stems (per plot), basal area (m² per plot), and spp richness refer to range (minimum and maximum) values. Stems and basal area are reported per plot.

*Number of stems, basal area and species richness were estimated for individual's ≥ 10 cm DBH for comparison.

terms of elevation and latitude in the models in order to describe unimodal trends. We used a Quasi-Poisson distribution for stem density and species richness since Poisson distribution showed overdispersion in both variables, and a Gaussian distribution with log-link function for basal area [55]. We calculated the percentage of variance explained with the following equation VE = (Null Deviance-Residual Deviance)/Null Deviance x 100.

To analyze patterns of species composition and floristic similarity, we used all data available from all plots (n = 491) and from all species fully identified (2341) with stems ≥10 cm DBH. To describe species composition, we applied a Non-Metric Multidimensional Scaling approach (NMDS) [56] based on a Bray-Curtis distance matrix [57] calculated from species abundance (stems per plots) between pairs of plots. We used a two-dimensional configuration because the final stress, an index of agreement between the distances in the graph configuration and the distances in the Bray-Curtis matrix, was 9.2 (most ecological community data sets have solutions with stress between 10 and 20; [58]. To explore the relationships between plots based on their species composition along elevation and latitudinal gradients we used Spearman's correlation coefficients [59] between the scores in the axes of the NMDS and elevation and latitude. To describe floristic similarity, we estimated Bray Curtis distance based on species abundance data [60]. Bray Curtis distances were calculated among pairs of plots within latitudinal bands. For this, we created eight latitudinal bands every 5° along the latitudinal gradient (from 8° N to 27° S) (Table 2). Floristic similarity (i.e. beta diversity) varies between 0 and 100%, i.e. values close to 0 imply less shared species. To explore the relationship between Bray Curtis distance with the geographical distance (km) and elevational difference (m asl) between pairs of plots we performed a linear mixed-effects model (LMM) [61] using latitudinal bands as a random factor. Geographical distances and elevational differences were obtained with Euclidean distances among pairs of plots within each latitudinal band, considering geographical coordinates and elevation, respectively. We calculated the proportion of variance explained within and among latitudinal bands, i.e., marginal and conditional $R^2$, respectively, according to the definitions given by Nakagawa & Schielzeth [62]. All analyses were performed in R 3.4.3 [63], using *AER* to test overdispersion in GLM, *vegan* for ordination analysis, and *lme4* for LMM.

## Results

The 491 plots were distributed along a 4138 km latitudinal gradient, encompassing a wide range of climates (Fig 3). Elevation ranges decreased from tropical to subtropical latitudes depending on the treeline location (Fig 3A). Mean annual air temperature and rainfall were

**Table 2. Summary of total species identified per latitudinal band, followed by mean Bray-Curtis distance (max value in brackets), mean geographical distance (max value in brackets), number of plots and mean plot size in hectares (range values in brackets) across the latitudinal gradient of the Andean Forest Network.**

| Latitudinal bands | Total # species identified | Mean Bray-Curtis distance (max) | Mean geographical distance (max) | # plots | Mean plot size (range) |
|---|---|---|---|---|---|
| 10° - 5° N | 729 | 0.05 (0.70) | 2.4 (6.3) | 19 | 0.90 (0.36 - 1) |
| 5° - 0° N | 310 | 0.08 (0.56) | 0.7 (1.5) | 23 | 0.22 (0.04 - 0.36) |
| 0° - 5° S | 1155 | 0.03 (0.97) | 2.1 (4.8) | 314 | 0.09 (0.01 - 1) |
| 5° - 10° S | 165 | 0.09 (0.71) | 0.3 (0.6) | 31 | 0.13 (0.01 - 1) |
| 10° - 15° S | 647 | 0.05 (0.72) | 1.5 (3.6) | 35 | 1 (1) |
| 15° - 20° S | - | - | - | 0 | - |
| 20° - 25° S | 151 | 0.16 (0.74) | 1.1 (2.6) | 47 | 1 (1) |
| 25° - 30° S | 53 | 0.13 (0.79) | 0.04 (0.08) | 22 | 0.88 (0.24 - 6) |

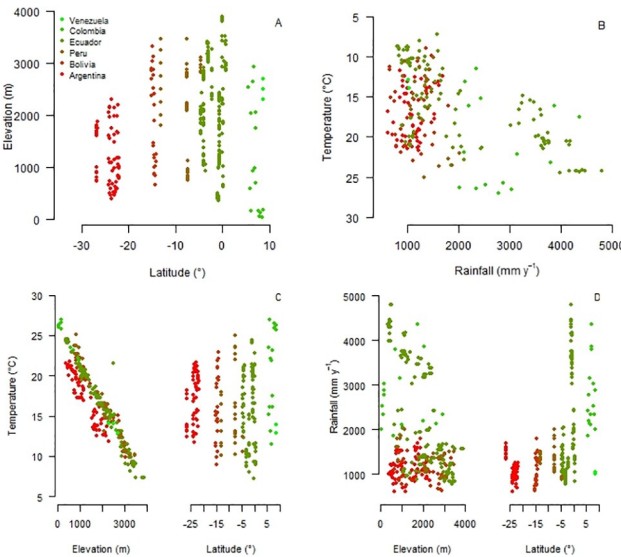

**Fig 3. Distribution of Andean Forest Network plots along climatic gradients.** Distribution of the Andean Forest Network plots along (A) spatial, and (B) climate gradients; and (C) mean annual temperature (inverted scale), and (D) annual rainfall along elevation and latitudinal gradients. Different colors refer to different countries.

positively correlated ($r = 0.46$, p < 0.001) (Fig 3B). As expected, mean annual temperature decreased with elevation ($r = -0.95$, p < 0.001) but did not vary with latitude ($r = -0.06$, p = 0.31) (Fig 3C). However, seasonal thermal amplitude increased with distance from the equator ($r = -0.93$, p < 0.001). Rainfall tended to decrease with elevation ($r = -0.38$, p < 0.001) and decreased towards higher latitudes ($r = 0.45$, p < 0.001) (Fig 3D).

We found that stem density tended to increase with elevation but decreased above 3000 m asl (S1 Appendix, Fig 4A). Stem density showed a unimodal trend with latitude, peaking at 10–15 degrees from the equator (Fig 4B). Elevation and latitude (and their quadratics terms) explained 43% of the variation in stem density, while plot size did not have a significant effect on stem density (S1 Appendix). Basal area tended to increase with elevation but peaked around 2000 m asl (Fig 4C). Basal area did not relate with latitude (Fig 4D). Plot size had a significant effect on basal area, small plots of 0.1-ha tended to present higher basal area than 1-ha plots (S1 Appendix). Combined, elevation, latitude and plot size explained 9.5% of the variation in basal area. Species richness showed a unimodal trend with elevation, peaking at 1000 m asl (Fig 4E) and decreasing with latitude being higher in the tropics (Fig 4F). Plot size also had a significant effect on species richness such that richness increased with the area sampled. Elevation, latitude and plot size together explained 70% of the variation in species richness. Taking into account the effect of plot size, we fitted a model only with 1-ha plots (n = 109) that showed, on average, higher species richness in tropical forest plots (Intercept = 68 tree species ha$^{-1}$, above 15° S) than in subtropical forest plots of Argentina (Intercept = 27 tree species ha$^{-1}$, between 22°-27° S) (Fig 4G, S1 Appendix).

Considering all plots (n = 491), we registered a total of 97,054 tree stems' $\geq$ 10 cm DBH, of which 90% (86,964) were identified to species level belonging to 2341 species, 584 genera and 133 botanical families (S2 Appendix), while 1095 remained unidentified as morphospecies of which 1019 remain to genera, 41 to family level, and 35 were not determined. Considering identified species (n = 2341), the richest genera were *Miconia* (105 species), followed by *Inga*

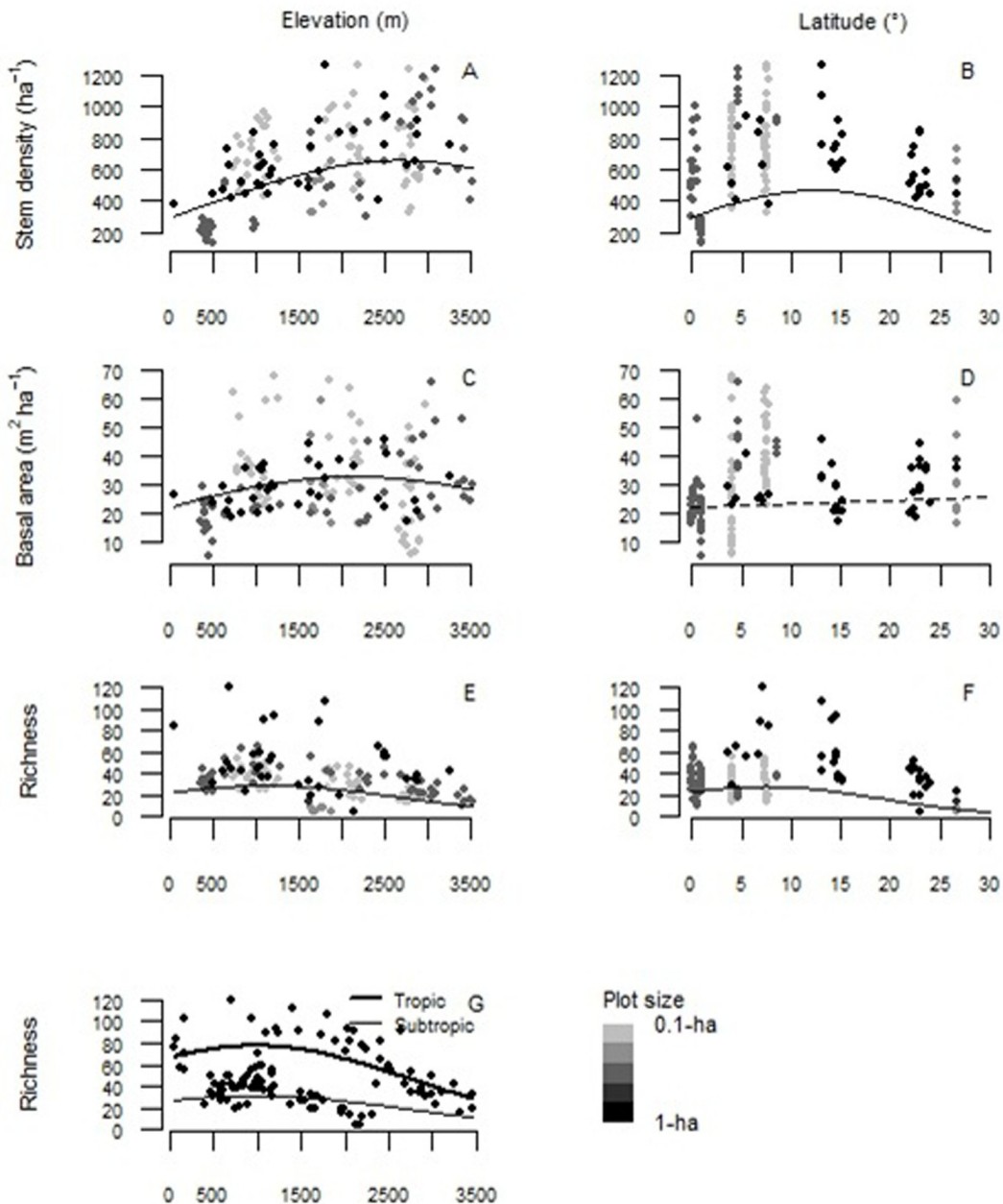

**Fig 4. Structural trends of Andean Forest Network plots along elevation and latitudinal gradients.** (A-B) Stem density, (C-D) basal area, and (E-G) species richness along Andean Forest Network plots in the elevation and latitudinal (expressed as distance in degrees to equator) gradient, respectively. Plots are differentiated by size with a gray gradient from 0.1-ha (60 plots), 0.24-ha (7 plots), 0.36-ha (49 plots), ~0.5-ha (6 of 0.40-ha and 1 of 0.48-ha) to 1-ha (109 plots of 1-ha, 2 of 0.96-ha, and 1 plot of 6-ha). Solid lines are significant (i.e. p < 0.001).

(54 species) and *Ocotea* (52 species); while the richest family was Fabaceae (177 species), followed by Melastomataceae (169 species), Rubiaceae (154) and Lauraceae (152 species). Only one species, *Myrsine coriacea* (Sw.) R.Br. ex Roem. & Schult, was shared among all data plots.

The study plots tended to group along two NMDS dimensions based on their species composition (Fig 5). Elevation and latitude correlated well with both NMDS axes (Axis 1: elevation

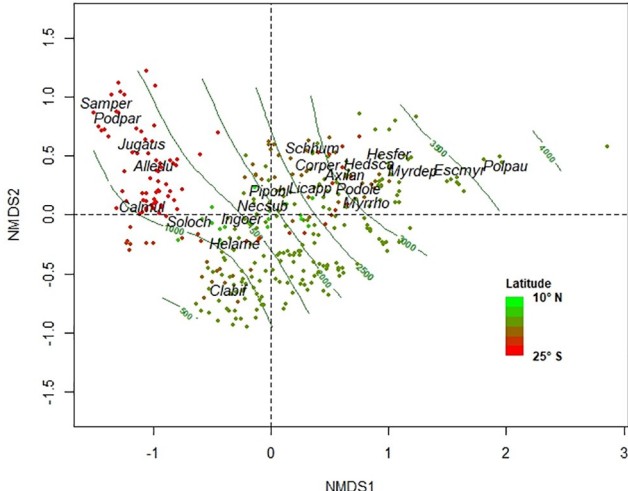

**Fig 5. Compositional trends of Andean Forest Network plots along elevation and latitudinal gradients.** Non-metric multidimensional scaling (NMDS) ordination diagram of Andean Forest Network plots based on tree species abundance. Different plots colors refer to different latitudes. Contours lines every 500 m asl are plotted. Species with highest abundance and frequency along NMDS 1 were plotted. Species code: Samper, *Sambucus peruviana*; Podpar, *Podocarpus parlatorei*; Jugaus, *Juglans australis*; Alledu, *Allophylus edulis*; Calmul, *Calycophyllum multiflorum*; Soloch, *Solanum ochrophyllum*; Helame, *Helicarpus americanus*; Clabif, *Clarisia biflora*; Ingoer, *Inga oerstediana*; Necsub, *Nectandra subbullata*; Pipobl, *Piper obliquum*; Licapp, *Licaria applanata*; Schhum, *Schefflera humboldtiana*; Corper, *Cornus peruviana*; Axilan, *Axinaea lanceolata*; Podole, *Podocarpus oleifolius*; Myrrho, *Myrcianthes rhopaloides*; Hesfer, *Hesperomeles ferruginea*; Hedsca, *Hedyosmum scabrum*; Myrdep, *Myrsine dependens*; Polpau, *Polylepis pauta*.

$r = 0.80$, and latitude $r = 0.45$; Axis 2: elevation $r = 0.51$, and latitude $r = -0.31$). Subtropical mountain forest plots between 22˚-27˚ S (Argentina) segregated from tropical forest plots along Axis 1 and were associated with species such as *Sambucus peruviana*, *Podocarpus parlatorei*, *Juglans australis*, *Allophylus edulis* and *Calycophyllum multiflorum*. Tropical premontane forest plots (< 1500 m asl) associated with *Solanum ochrophyllum*, *Clarisia biflora* and *Heliocarpus americanus*, while *Nectandra subbullata* and *Piper obliquum* related to tropical lower montane forest plots (i.e. 1500 to 2700 m asl). *Licaria applanata*, *Cornus peruviana*, *Podocarpus oleifolius*, *Myrcianthes rhopaloides*, *Hedyosmum scabrum*, *Hesperomeles ferruginea* and *Myrsine dependens* were associated with tropical upper forest plots (i.e. 2700 to 3500 m asl) while *Polylepis pauta* grouped with plots above upper forest line (>3500 m asl).

Bray-Curtis distance (i.e., floristic similarity) across all Andean plots was on average 4% (the lower the value the less similarity among plots). Latitudinal bands between 10˚ N and 15˚ S (i.e. tropical montane forest plots) showed mean values between 3 to 9% in floristic similarity while in latitudinal bands below 20˚ S (i.e. subtropical montane forest plots), floristic similarity presented mean values of 13 and 16% (Table 2). These latitudinal bands explained 30.7% of floristic similarity variation among latitudinal bands (i.e., tropical vs. subtropical forest plots) while geographical and elevational distance between plots explained only 2% of floristic similarity decay within latitudinal bands (Fig 6A and 6B; S3 Appendix). Together, geographical distance and elevational difference between pairs of plots within and among latitudinal bands explained 32.8%.

## Discussion and conclusions

The 491 forest plots considered in this study covered broad range of latitudes, elevations and environmental conditions. As a general pattern, temperature and rainfall decreased with

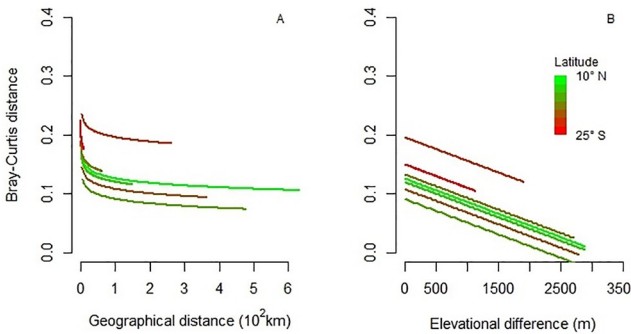

**Fig 6. Floristic similarity of Andean Forest Network plots within latitudinal bands.** Bray-Curtis distance (i.e. floristic similarity) between pairs of plots of the Andean Forest Network as function of geographical distance (A), and elevation difference (B) within latitudinal bands.

elevation, implying that forest plots located at lower elevations were subject to warmer and wetter conditions. While this pattern is well established for temperature, it was different from what Urrutia et al. [5] reported for the Andes where rainfall did not co-vary linearly with elevation. In the latitudinal gradient, rainfall increased towards the equator while mean temperature did not show a clear pattern but seasonal thermal amplitude was higher with increasing latitude (i.e. further from the equator) [64].

Both stem density and basal area were related with the gradients covered by our dataset. However, patterns of stem density were more consistent with elevation and latitude than for basal area. Forest plots with higher stem density (and basal area) were located at higher elevations (> 2000 m asl) and towards the equator. Stem density tended to peaked at 10-15° latitude (e.g. plots of Colombia). The observed increase in the mean values of structural variables with elevation has not been frequently reported; on the contrary, some authors reported the opposite pattern of decreasing density [32], basal area or productivity with elevation [30, 33]. Probably, there are other variables masked in our study, such as the latitudinal gradient (because the highest densest plots are in the tropics) or maybe some local climate associated to topographic conditions (i.e. wetter or drier slopes, soil properties). In addition land use change history may influence tree density as well in the case of secondary forests, i.e. if associated with tree invasive species [65]. Furthermore, the use of small plots (i.e. 0.1-ha) which represent about 12% of plots in AFN, tended to overestimate basal area in comparison to larger plots (mean basal area per hectare 33.9± 1.9 for 0.1-ha plots; and 26.5± 0.7 for 1-ha plots). Moreover, the basal area of these small plots presented the opposite pattern to the general trend, i.e. decreasing basal area with elevation, being consistent with other studies that use small plots [30, 33]. A stronger edge effect and differences in disturbance intensities (disturbed vs. mature forests) as well as majestic forest effect could be expected in these small plots. In addition, smaller plots (e.g. between 0.01 and 0.05-ha), which were discarded from our main analyses, showed a very large variability and dispersion when scaled to 1-ha (e.g. 0.01 ha plot with 1 stem would extrapolated to 100 stems ha$^{-1}$, and in a 0.04-ha plot with 35 trees extrapolated value of basal area would be as high as 131.2 m$^2$ ha$^{-1}$). Overall, plots ~ 0.5-ha or larger may be preferred for describing patterns at regional scales.

The greater species richness at lower latitudes compared to the higher latitudes has been widely studied. Explanations for this pattern include the Janzen and Connell conspecific density dependent hypothesis which maintain diversity in plant communities by reducing survival rates of conspecific seedlings located close to reproductive adults or in areas of high conspecific density [66], the narrow environmental niche [67], the recruitment limitation hypothesis [68].

Also, the tropical niche conservatism hypothesis may explain the observed pattern where species diversity and biogeographic processes are driven by historical climate where the stable, wet, warm and a-seasonal tropical climate promote high diversity [69, 70, 71] and narrower niches (i.e. the latitudinal gradient hypothesis) [64, 72].

Subtropical forest plots (i.e., plots in Argentina) clearly segregated from the rest of the tropical forest plots based on species composition, although the elevation gradient was also important, particularly in tropical plots. In terms of floristic similarity (i.e., beta diversity or species turnover) tropical forest plots showed between 3% to 9% of species shared implying high species turnover among their plots within latitudinal bands, while subtropical plots shared more species, between 13% and 16%, implying lower species turnover among plots associated with low diversity of tree species. The high species turnover found in the tropics was reported by several authors and attributed to the combination of climatic variables [73], to multiple coexistence mechanisms [74], and to the high biotic interaction found at these lower latitudes [75] as well as high habitat heterogeneity driven by ample environmental gradients [43, 76].

As well, tropical forest plots separated by larger geographical distances (km) may be less similar due the presence of intermontane valleys and the influence of different floristic regions. This pattern (i.e. less floristic similarity or high species turnover found in the tropics) has also been reported for Andean high alpine vegetation [77]. As the variation explained among latitudinal bands in this regional context was very prominent, geographical and elevational distances between forest plots seemed to be relatively less important. Considering the elevation gradient of the Andes, our study suggests a mid-elevation peak in species diversity for both tropics and subtropics which respond to the mid domain effect hypotheses [78] related to mountain topographic constraints as a result of low temperatures although the mechanisms involved are not clear [36]. Our findings may imply that forests located in tropical and towards mid elevations sections of the Andes have the potential to accumulate more biomass, and consequently sequestrate more carbon, than forests located away from the equator or towards lower elevations. Conservation efforts may be particularly important within this forests.

The creation of the Andean Forest Network has served as an important platform for communication where researchers were able to discuss the use and interpretation of forest plot information in a context of global environmental change. The Andean Forest Network expects to continue expanding its collaborative work across the region adding forest plots along the Andes. Some of the key issues that emerged from its functioning includes: 1) strengthening the institutional arrangements within the Andean Forest Network, and between it and other actors interested in conducting research in Andean forests; 2) implementing mechanisms to promote the long-term sustainability of the Network and the monitoring activities of its members. This will require identifying funding opportunities within the region and abroad, from diverse sources such as climate funding, grants for field research, public funding for science, among others. Also, the compilation and management of the large database includes the need of: 1) improving species identifications and, when possible, standardizing morphospecies along plots; 2) developing a "decentralized" database or an online platform to share data quickly and easily with the least possible errors; 3) strengthening the on-site collection of climate data from meteorological stations; 4) filling current geographical gaps (for example, Bolivian montane dry forest or central Peruvian yungas) and encouraging institutions/researchers to contribute with their forests plot data to the Network, increasing the area covered per country and promoting re-measurement of the installed plots; 5) promoting the implementation of other modules available in protocols (e.g. carbon stocks, species functional traits) [47] and other useful complementary measurements (e.g. dendrometer bands) in all the permanent forest plots of the Network; 6) detecting situations that are locally important; 7) promoting comparative research for monitoring the impacts of anthropogenic activities across the region, including

the dynamics of invasive species (i.e. the case of *Ligustrum lucidum* in Argentina) [79], or primary/secondary succession as a result of volcanism, forest fires, hurricanes, or land use change (abandonment of agricultural and livestock lands); 8) exploring in detail the association between changes in permanent plots and changes in remotely sensed descriptors of functioning (e.g., NDVI) [80], promoting joint research to develop high resolution models of climate change for the Andean region, and descriptions of land use change [12].

Overall, this study is one of the first attempts to integrate forest structure and composition in the Andes showing clear patterns along its elevation and latitudinal gradients. In this sense, the AFP Network constitutes an important initiative to fill geographical gaps in regions such as the tropical and subtropical Andes.

.

## Supporting information

**S1 Table. Andean Forest Network extended metadata.** [a]Country codes are AR, Argentina; BO, Bolivia; PE, Peru; EC, Ecuador; CO, Colombia; VE, Venezuela. [c]Plot shape refer to R, rectangular; Q, quadrate, and I, irregular; and dimensions refer to length (L) and width (W); [d]Mean annual rainfall and temperature derived from Chelsa, [e]Time since last disturbance at plot establishment. Number of stems and basal area were extrapolated to 1-ha. [*]Number of stems, basal area and species richness were estimated for individual's $\geq 10$ cm DBH for comparison.
(DOCX)

**S1 Appendix. Generalized linear models.** Summary of Generalized Linear Models (GLM) for stem density, basal area and species richness considering elevation, latitude and plot size as explanatory variables. SE = Standard error. VE = Variance explained ((Null Deviance—Residual Deviance)/Null Deviance x 100). ([*]) Only 1-ha plots were included in the model considering Tropical and Subtropical Andes.
(DOCX)

**S2 Appendix. Abundance of tree species per country.** Abundance of tree species per country, considering individuals $\geq 10$ cm DBH listed by botanical family. Species and family names were actualized with TROPICOS in September 2019 (http://www.tropicos.org).[a]Country codes are AR, Argentina; BO, Bolivia; PE, Peru; EC, Ecuador; CO, Colombia; and VE, Venezuela.
(DOCX)

**S3 Appendix. Linear mixed-effects model.** Summary of Linear Mixed-effects Model (LMM) for Bray Curtis distance between pairs of forest plots, varying intercept by latitudinal bands every 5˚. Final model fitted by REML. SD = Standard deviation, VE = Variance explained (i.e. marginal and conditional $R^2$).
(DOCX)

## Acknowledgments

We thank all the people involved in the installation and monitoring of plots along the Andean Forest Network, and to the different institutions and financiers in different countries who made the field work possible. We also thank all the taxonomic experts, students and local guides that were involved in the collection of the field data and in the identification of plant specimens. We also thank Parque Sierra de San Javier, Administración de Parques Nacionales and private owners in Argentina for facilitating the establishment and monitoring of permanent

plots, the Instituto Nacional de Parques in Venezuela for facilitating establishment of permanent plots in the Sierra Nevada National Park. We thank Andrea Izquierdo, Javier Foguet and Silvia Pacheco for their contribution with climatic data and collaboration with the map of Fig 2.

## Author Contributions

**Conceptualization:** Agustina Malizia, Cecilia Blundo, Julieta Carilla, Oriana Osinaga Acosta, Francisco Cuesta, Alvaro Duque.

**Data curation:** Oriana Osinaga Acosta, Francisco Cuesta, Alfredo Fernando Fuentes, Luis E. Gámez Álvarez.

**Formal analysis:** Cecilia Blundo.

**Funding acquisition:** Agustina Malizia, Cecilia Blundo, Francisco Cuesta, Lucio Malizia, Manuel Peralvo.

**Investigation:** Agustina Malizia, Cecilia Blundo, Julieta Carilla, Oriana Osinaga Acosta, Francisco Cuesta, Alvaro Duque, Nikolay Aguirre, Zhofre Aguirre, Michele Ataroff, Selene Baez, Marco Calderón-Loor, Leslie Cayola, Luis Cayuela, Sergio Ceballos, Hugo Cedillo, William Farfán Ríos, Kenneth J. Feeley, Alfredo Fernando Fuentes, Luis E. Gámez Álvarez, Ricardo Grau, Juergen Homeier, Oswaldo Jadan, Luis Daniel Llambi, María Isabel Loza Rivera, Manuel J. Macía, Yadvinder Malhi, Lucio Malizia, Manuel Peralvo, Esteban Pinto, Sebastián Tello, Miles Silman, Kenneth R. Young.

**Methodology:** Cecilia Blundo, Julieta Carilla.

**Project administration:** Francisco Cuesta, Manuel Peralvo.

**Supervision:** Agustina Malizia, Francisco Cuesta.

**Visualization:** Agustina Malizia, Cecilia Blundo, Julieta Carilla, Oriana Osinaga Acosta.

**Writing – original draft:** Agustina Malizia, Cecilia Blundo, Julieta Carilla, Oriana Osinaga Acosta, Francisco Cuesta.

**Writing – review & editing:** Agustina Malizia, Cecilia Blundo, Julieta Carilla, Oriana Osinaga Acosta, Francisco Cuesta, Alvaro Duque, Marco Calderón-Loor, Leslie Cayola, Luis Cayuela, Kenneth J. Feeley, Ricardo Grau, Juergen Homeier, Luis Daniel Llambi, María Isabel Loza Rivera, Manuel J. Macía, Lucio Malizia, Manuel Peralvo, Esteban Pinto, Kenneth R. Young.

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
