## [Decision Letter · Decision Letter 0]

24 Dec 2019

PONE-D-19-30260

Structural and compositional trends along elevation and latitude gradients of Andean forests

PLOS ONE

Dear Mrs Malizia,

Thank you for submitting your manuscript to PLOS ONE. After careful consideration, we feel that it has merit but does not fully meet PLOS ONE’s publication criteria as it currently stands. Therefore, we invite you to submit a revised version of the manuscript that addresses the points raised during the review process.

ACADEMIC EDITOR: 

Please seriouly consider the concerns of the referees and make revisions according to their suggsetions,especially you should pay attention to comments about data availability(this is also clearly required by the journal) and the discusssions on the major results.

We would appreciate receiving your revised manuscript by Feb 07 2020 11:59PM. To enhance the reproducibility of your results, we recommend that if applicable you deposit your laboratory protocols in protocols.io, where a protocol can be assigned its own identifier (DOI) such that it can be cited independently in the future. For instructions see: http://journals.plos.org/plosone/s/submission-guidelines#loc-laboratory-protocols

We look forward to receiving your revised manuscript.

Kind regards,

RunGuo Zang

Academic Editor

PLOS ONE

Journal Requirements:

Additional Editor Comments:

Please integrate the concerns of the referees and make improvement accordingly

Reviewers' comments:

Reviewer's Responses to Questions

**Comments to the Author**

1. Is the manuscript technically sound, and do the data support the conclusions?

Reviewer #1: Yes

Reviewer #2: Partly

2. Has the statistical analysis been performed appropriately and rigorously? 

Reviewer #1: I Don't Know

Reviewer #2: Yes

3. Have the authors made all data underlying the findings in their manuscript fully available?

Reviewer #1: No

Reviewer #2: Yes

4. Is the manuscript presented in an intelligible fashion and written in standard English?

Reviewer #1: Yes

Reviewer #2: Yes

5. Review Comments to the Author

Reviewer #1: General remarks:

This paper reports on trends in stem number, basal area and species diversity patterns for trees in a series of plots spanning ~4000 km of latitude and 4000 m of elevation range. The results are a useful addition to the analysis of trends in the physical and biological structure of Andean forests. In general, I found the manuscript clearly written and not over-reaching in its conclusions.

I believe that most last contribution of this manuscript would be the publication of the data upon which all the analyses were conducted. Unfortunately, it appears that this manuscript does not meet PLOS One standards of data availability. The authors state that “All relevant data are within the manuscript and its Supporting Information files.”, and that “We used the Andean Forest 70 Network (Red de Bosques Andinos, www.redbosques.condesan.org/) database which, at 71 present, includes 491 forest plots (totaling 156.3 ha, ranging from 0.01 to 6 ha) representing 72 a total of 86,964 tree stems ≥ 10 cm diameter at breast height belonging to 2341 identified 73 species, 584 genera and 133 botanical families.”

This is a manuscript analyzing Andean forest tree structure and diversity. The data used are what the authors describe above, i.e. size, location and identification data on 86,964 tree stems. It appears that the data on these individual stems are not, contrary to what the authors state, presented in either the manuscript or in the Supporting Information. I followed the link to the Red de Bosques Andinos and was not able to find any of the individual stem data. I did find a link to “datos” (data in Spanish), but that led only to plot metadata, not to the actual data.

In summary, even with a diligent effort I was unable to find the data on which this manuscript is based. If somehow the stem data are in fact publicly accessible without restriction somewhere, they are not easily findable even with reasonable effort. The authors should clearly show exactly where the individual stem data can be freely accessed with no restrictions.

The authors state “The plot data that support the findings of this study are available from Andean 251 Forests Network upon reasonable request”, so it appears to me that the data are not publicly available without restriction. PLoS ONE’s data availability policy states: “PLoS journals require authors to make all data necessary to replicate their study’s findings publicly available without restriction at the time of publication.” To replicate the study’s findings requires access to the individual stem data. Unless public access without restriction is provided the manuscript is not acceptable according to PLoS ONE’s standards.

In the case of this manuscript complying with PLoS ONE’s data policy is not an onerous requirement. Stem data for 87,000 individuals is not large dataset and the data could easily be easily be included as a table in the Supporting Information. An example from PLoS One is Clark et al. 2017 https://journals.plos.org/plosone/article?id=10.1371/journal.pone.0183819#sec013. In their S1 Table those authors provide individual stem data for size, location and taxonomic identity of ~80,000 individual census measurements, which is a similar-size data set. I strongly encourage the authors to include the individual stem data as a table in Supporting Information so that the exact version of the dataset that they used to generate the paper is archived with the paper. Data in a dataset of this nature are constantly and very correctly changing as errors are detected and corrected and as taxonomic concepts evolve. It would be difficult in the future to recreate these analyses unless the version of the database at the time of these analyses is published. Such publication of the freely available base data is a PLoS ONE requirement for publication, and it is also best scientific practice.

My pdf copy of the manuscript lost line numbering after line 274, so after that I use the page numbers of my pdf copy to reference comments.

It is unclear how many individuals were analyzed. The Abstract states 86,944 stems, but on page 26 a total of 97,054 stems is mentioned. It is not clear if the 86,944 applies only to the analyses involving taxonomic identity. I assume that the total 97,054 stems were used in the analyses of basal area and stem number, there is no reason to discard individuals for these analyses. In any case the sample size for each set of analyses must be clarified.

The authors are very aware of the issues of plot size and its effects on scaling to larger spatial scales. They are clear that they omitted many very small plots (<0.1 ha) from their analyses. However the analyzed plots are located in mountainous terrain, and are likely spatially biased towards rare flat areas. In addition, most (all?) of the plots were subjectively sited (not sited using a spatially random protocol) and may be subject to the “majestic forest” bias. It would be useful for the authors to provide their opinion of the effects on non-random plot placements on these results.

Other comments:

I suggested deleting lines 230-241, this section is not directly relevant to the reported research.

Table 1. Clarify if temperature and rainfall are annual averages. This table would be easier to read with vertical lines separating the variables, and “Min” and “Max” over each of those columns.

Page 22. Spell out CHELSA. Rapid variation in environmental gradients is well known in tropical mountain area. Are there any site-specific data to check the accuracy of the predicted climate with actual climate on the ground?

The whole paragraph on page 22 on climate data reads like climate was actually measured on the ground. In fact climate variables were predicted from remotely sensed data, with an unknown (or unreported) degree of uncertainty. All of these data should be labeled as “estimated” or “predicted” to make clear that actual ground-measured data are not being reported. It’s fine to report remotely sensed data, but the manuscript should be clear that is what’s being reported. Based on this manuscript there is no way to know how good these estimates are at the plot level. If the authors in fact do know the accuracy of these predictions it would be useful to report that.

Figure 3 caption, B, change “climate” to “estimated annual rainfall”. What is the reason for the inverted scale on panel B? All the other scales are linear increasing, so this exception is confusing. I suggest plotting plot B like the other variables, both variables increasing from the origin.

Figure 4. Move Elevation and Latitude to the top of each column rather than below (it’s confusing as presented since the columns are of unequal length).

Table 2. Add units to the column headings were these are missing.

Figure 4 is confusing. What is the difference between panels E and G? The text states that “Species richness decreased with latitude (Fig 4E-F)” but panel E is based on elevation, not latitude. Please clarify in the text and in the figure legend.

Consider replacing “hump-shaped elevational pattern of tree species richness” with “mid-elevation peak in species diversity”.

Reviewer #2: I liked the manuscript and the overall approach. I also enjoy reading synthesis-based analyses that are based on field-plot data. I also think this particular plot network has an enormous potential for future work and research to support conservation and management of Andean forests. However, in its current form, this work requires more analytical thinking, especially to discuss the main findings and the causal mechanisms behind the patterns that were reported. See the attached file for detailed comments.

6. PLOS authors have the option to publish the peer review history of their article (what does this mean?). If published, this will include your full peer review and any attached files.

Reviewer #1: Yes: David B. Clark

Reviewer #2: No

---

## [Author Response · Author response to Decision Letter 0]

16 Mar 2020

March 30th, 2020

Dr. Joerg Heber, Editor-in-Chief

PLoS ONE

Dear Dr. Herber,

I am writing to submit our revised manuscript (PONE-D-19-30260) entitled Elevation and latitude drives structure and tree species composition in Andean forests: results from a large-scale plot Network for consideration for publication in PLoS ONE. We are very thankful for your comments and those of the reviewers as they have allowed us to improve our manuscript. We have carefully considered each of them and addressed all of them. In this sense, we have made clear the point about data availability as well as we have improved the discussion concerning the major results. Please find below the detailed responses to each comment. Line and Page numbers referred correspond to the revised manuscript with track changes. 

With best regards,

Dra. Agustina Malizia (On behalf of all authors)

Instituto de Ecología Regional (IER) (CONICET-UNT)

Editor Comments

Please seriously consider the concerns of the referees and make revisions according to their suggestions, especially you should pay attention to comments about data availability (this is also clearly required by the journal) and the discussions on the major results. 

Response: We have paid attention to all comments made by both reviewers, incorporated all of them in the manuscript, and responded to each one in detailed. Please, see below. 

Reviewers' comments:

Reviewer's Responses to Questions

Comments to the Author

1. Is the manuscript technically sound, and do the data support the conclusions?

Reviewer #1: Yes

Reviewer #2: Partly ________________________________________

2. Has the statistical analysis been performed appropriately and rigorously?

Reviewer #1: I Don't Know

Reviewer #2: Yes________________________________________3. Have the authors made all data underlying the findings in their manuscript fully available?

Reviewer #1: No

Reviewer #2: Yes

4. Is the manuscript presented in an intelligible fashion and written in Standard English?

Reviewer #1: Yes

Reviewer #2: Yes ________________________________________

5. Review Comments to the Author

Reviewer #1: 

General remarks:

This paper reports on trends in stem number, basal area and species diversity patterns for trees in a series of plots spanning ~4000 km of latitude and 4000 m of elevation range. The results are a useful addition to the analysis of trends in the physical and biological structure of Andean forests. In general, I found the manuscript clearly written and not over-reaching in its conclusions.

I believe that most last contribution of this manuscript would be the publication of the data upon which all the analyses were conducted. Unfortunately, it appears that this manuscript does not meet PLOS One standards of data availability. The authors state that “All relevant data are within the manuscript and its Supporting Information files.”, and that “We used the Andean Forest Network (Red de Bosques Andinos, www.redbosques.condesan.org/) database which, at present, includes 491 forest plots (totaling 156.3 ha, ranging from 0.01 to 6 ha) representing a total of 86,964 tree stems ≥ 10 cm diameter at breast height belonging to 2341 identified species, 584 genera and 133 botanical families.”

This is a manuscript analyzing Andean forest tree structure and diversity. The data used are what the authors describe above, i.e. size, location and identification data on 86,964 tree stems. It appears that the data on these individual stems are not, contrary to what the authors state, presented in either the manuscript or in the Supporting Information. I followed the link to the Red de Bosques Andinos and was not able to find any of the individual stem data. I did find a link to “datos” (data in Spanish), but that led only to plot metadata, not to the actual data.

In summary, even with a diligent effort I was unable to find the data on which this manuscript is based. If somehow the stem data are in fact publicly accessible without restriction somewhere, they are not easily findable even with reasonable effort. The authors should clearly show exactly where the individual stem data can be freely accessed with no restrictions.

Response: In Table S1 we included: i) number of stems, ii) basal area (both extrapolated to 1-ha) and iii) species richness for each of the 491 plots. Additionally, in Appendix 1 we reported the abundance of tree species per country. This information constitutes the primary input used in the analyses to describe the main structural and compositional trends across the Andean forests. Thus, we included the following text in the manuscript to make explicitly clear that the information upon which replicate the analyses is available in the supporting information: “All relevant data upon which all the presented results in this manuscript are based on is included in its Supporting Information files…”. (P13, L296-300). We also clarified that the link to Red de Bosques Andinos (https://redbosques.condesan.org/) is to check general information about the Network (P11, L230-231). 

The authors state “The plot data that support the findings of this study are available from Andean Forests Network upon reasonable request”, so it appears to me that the data are not publicly available without restriction. PLoS ONE’s data availability policy states: “PLoS journals require authors to make all data necessary to replicate their study’s findings publicly available without restriction at the time of publication.” To replicate the study’s findings requires access to the individual stem data. Unless public access without restriction is provided the manuscript is not acceptable according to PLoS ONE’s standards. 

Response: In order to replicate the study, the information needed is the following: i) number of stems, ii) basal area and iii) species richness for each of the 491 plots which, and as we stated previously, it is provided in the supporting files. To avoid confusion, we have deleted the sentence: “The plot data that support the findings of this study are available from Andean Forests Network upon reasonable request”.

In the case of this manuscript complying with PLoS ONE’s data policy is not an onerous requirement. Stem data for 87,000 individuals is not large dataset and the data could easily be included as a table in the Supporting Information. An example from PLoS One is Clark et al. 2017 https://journals.plos.org/plosone/article?id=10.1371/journal.pone.0183819#sec013. In their S1 Table those authors provide individual stem data for size, location and taxonomic identity of ~80,000 individual census measurements, which is a similar-size data set. I strongly encourage the authors to include the individual stem data as a table in Supporting Information so that the exact version of the dataset that they used to generate the paper is archived with the paper. Data in a dataset of this nature are constantly and very correctly changing as errors are detected and corrected and as taxonomic concepts evolve. It would be difficult in the future to recreate these analyses unless the version of the database at the time of these analyses is published. Such publication of the freely available base data is a PLoS ONE requirement for publication, and it is also best scientific practice.

Response: We understand the reviewer's request; however, as explained above, the data required to replicate this research is available in the supporting information included in the manuscript. Also, the Andean Forest Network dataset belongs to the different research groups that are part of the Network. Currently, these research groups are implementing projects and research with master's and doctoral students based in part on the same information presented here. It is not our intention to hinder research initiatives that require an embargo of the data until the different research projects are complete, making the base information available.

My pdf copy of the manuscript lost line numbering after line 274, so after that I use the page numbers of my pdf copy to reference comments.

Response: We added line numbers until the end of the manuscript. Apparently, lines were missing after number 274 as the reviewer stated. We apologize for the inconveniences.

It is unclear how many individuals were analyzed. The Abstract states 86,964 stems, but on page 26 a total of 97,054 stems is mentioned. It is not clear if the 86,964 applies only to the analyses involving taxonomic identity. I assume that the total 97,054 stems were used in the analyses of basal area and stem number, there is no reason to discard individuals for these analyses. In any case the sample size for each set of analyses must be clarified.

Response: Corrected. We added a clarifying sentence: For calculations of stem density and basal area we used all stems ≥10 cm DBH (n = 97,054) while for species richness we used stems ≥10 cm identified to species level (n = 86,964) (P16, L337-339). We also clarified this in the abstract: “…representing a total of 86,964 identified stems ≥ 10 cm diameter…” (P4, L74-75).

The authors are very aware of the issues of plot size and its effects on scaling to larger spatial scales. They are clear that they omitted many very small plots (<0.1 ha) from their analyses. However the analyzed plots are located in mountainous terrain, and are likely spatially biased towards rare flat areas. In addition, most (all?) of the plots were subjectively sited (not sited using a spatially random protocol) and may be subject to the “majestic forest” bias. It would be useful for the authors to provide their opinion of the effects on non-random plot placements on these results.

Response: Forest plots may not have been located strictly at random thus is challenging to guarantee that some of them were not subject to the majestic forest bias. However, forest plots setup were based on the combination of several factors: (1) accessibility, (2) sustainable over time to ensure long term monitoring for some plots, and (3) homogeneity in terms of forest type, topography, and disturbances in order to be representative of particular forest types. We clarified this in text (P13, L275-278). In this sense, we have stated that: “The majority of the plots are placed in mature forests, but the Network also includes several older secondary forests plots, i.e. > 30 years old as a result of human activities and the associated land-use changes” (P13 279-281).

Concerning the majestic forest bias, small plots may be subjected to this effect showing higher estimated values of basal area (Phillips et al. 2002). In this sense, in the result section we have reported that “Plot size had a significant effect on basal area, small plots of 0.1-ha tended to present higher basal area than 1-ha plots (Appendix 1) (P19 L396-397). However, taking into account the three variables combined, the variance explain is less than 10%, as we have stated: “Combined, elevation, latitude and plot size explained 9.5% of the variation in basal area” (P19 L398). Finally, in the discussion section we added: “…as well as majestic forest effect could be expected in these small plots (P24 L504-505). 

Phillips OL, Malhi Y, Vinceti B, Baker T, Lewis SL, Higuchi N, ... & Ferreira LV. (2002). Changes in growth of tropical forests: evaluating potential biases. Ecological Applications, 12(2), 576-587.

Other comments:

I suggested deleting lines 230-241, this section is not directly relevant to the reported research.

Response: Agreed, we deleted it. 

Table 1. Clarify if temperature and rainfall are annual averages. This table would be easier to read with vertical lines separating the variables, and “Min” and “Max” over each of those columns.

Response: Done. We clarify in the legend of Table 1 (P15, L312) that we referred to total annual rainfall and mean annual temperature. Also, we added vertical lines in this table, and “min” and “max” as suggested. Additionally, in legend of Table S1 we also clarified that we referred to total annual rainfall and mean annual temperature. 

Page 22. Spell out CHELSA. Rapid variation in environmental gradients is well known in tropical mountain area. Are there any site-specific data to check the accuracy of the predicted climate with actual climate on the ground?

Response: We spelled out CHELSA as “Climatologies at high resolution for the Earth’s land surface areas” (P16 L318). We were not able to corroborate the accuracy of the predicted climate with actual climate on the ground. In this sense, Andean precipitation patterns are currently not yet well described, due to the high spatio-temporal variability and low density of rain gauges, thus we expect some underestimations and overestimations in precipitation values (Urrutia and Vuille 2009; Buytaert et al. 2010). Nevertheless, we believe we have used the best available climatic information to describe general climatic patterns across Andean forest plots. 

Buytaert, W., Vuille, M., Dewulf, A., Urrutia, R., Karmalkar, A. & Célleri, R. (2010) Uncertainties in climate change projections and regional downscaling in the tropical Andes: implications for water resources management. Hydrol. Earth Syst. Sci., 14, 1247-1258

The whole paragraph on page 22 on climate data reads like climate was actually measured on the ground. In fact climate variables were predicted from remotely sensed data, with an unknown (or unreported) degree of uncertainty. All of these data should be labeled as “estimated” or “predicted” to make clear that actual ground-measured data are not being reported. It’s fine to report remotely sensed data, but the manuscript should be clear that is what’s being reported. Based on this manuscript there is no way to know how good these estimates are at the plot level. If the authors in fact do know the accuracy of these predictions it would be useful to report that.

Response: We rephrased the information within this paragraph adding the word estimated and predicted in order to make clear that it was remotely sensed data information (P16, L317-324). 

Figure 3 caption, B, change “climate” to “estimated annual rainfall”. What is the reason for the inverted scale on panel B? All the other scales are linear increasing, so this exception is confusing. I suggest plotting plot B like the other variables, both variables increasing from the origin. 

Response: We showed the inverted scale on Fig 3B (y axis = temperature) to emphasize illustratively the location of plots associated with elevation and temperature, i.e. higher plots are also the colder ones and are graphed in similar position in both panels (Fig 3A, B). We also clarified it (i.e. inverted scale) in the legend of Fig 3 (P19, L387).

Figure 4. Move Elevation and Latitude to the top of each column rather than below (it’s confusing as presented since the columns are of unequal length). 

Response: Done. We moved Axis X (Elevation and Latitude) above to the top of each column

Table 2. Add units to the column headings were these are missing. 

Response: Done. We added the symbol “#” (number) in two of the columns. 

Figure 4 is confusing. What is the difference between panels E and G? The text states that “Species richness decreased with latitude (Fig 4E-F)” but panel E is based on elevation, not latitude. Please clarify in the text and in the figure legend.

Response: Done. We clarified that Fig 4F is the one that refers to latitude (P19, L399-400). Fig 4E does not distinguish between tropical and subtropical sections, while Fig 4G does distinguish between tropics and subtropics taking into account only 1ha-plots (n = 109). This is stated in text (P19, 403-406).

Consider replacing “hump-shaped elevational pattern of tree species richness” with “mid-elevation peak in species diversity”. 

Response: Done. We replaced as suggested.

6. PLOS authors have the option to publish the peer review history of their article (what does this mean?). If published, this will include your full peer review and any attached files.

Do you want your identity to be public for this peer review? For information about this choice, including consent withdrawal, please see our Privacy Policy.

Reviewer #1: Yes: David B. Clark

Reviewer #2: No

Reviewer #2: I liked the manuscript and the overall approach. I also enjoy reading synthesis-based analyses that are based on field-plot data. I also think this particular plot network has an enormous potential for future work and research to support conservation and management of Andean forests. However, in its current form, this work requires more analytical thinking, especially to discuss the main findings and the causal mechanisms behind the patterns that were reported. See the attached file for detailed comments.

Manuscript Review PLOS ONE (PONE-D-19-30260)

Structural and compositional trends along elevation and latitude gradients of Andean forests By Malizia et al…

General comments:

This study is based on data collected from an impressive number of research sites, between temporary and permanent forest plots, located in different parts of the Andes in South America, an important biological hotspot that has received less attention compared to lowland forests. Broadly, this paper takes advantage of the wide and complex environmental range covered by the dataset to address the questions of the combined effects of elevation and latitude on the structure and tree species composition of Andean forests. By means of generalized models, the authors found that forest density (i.e. number of trees or stems per unit of area) increased with elevation with an opposite effect from latitude. The differences in species richness found between tropical and subtropical Andes are not surprising and are aligned with previous studies. However, the degree of similarity (or dissimilarity) in terms of species diversity in these two broad regions within the Andes was an interesting finding. The authors very well highlight the immense value that this Andean plot-network (i.e. Red de Bosques Andinos) has for strengthening scientific research in the region and promote collaboration. These are not ‘novel’ questions but given the amount of data used, especially compared to an earlier study from the same group (Baez et al. 2015 PLoS ONE), I find the paper of high interest for many scientists and forest ecologists, and readers of PLoS ONE. However, in its current form this manuscript remains largely descriptive and a better explanation of the causal mechanisms behind how elevation and/or latitude drives structure and composition is missing. The implications of the results to other aspects of forest structure (e.g. carbon) or dynamics (e.g. stem turnover) could be added too. I think the paper should be improved, especially in the discussion section, before being considered for publication. I am pointing out several comments, questions, and occasionally, a few suggestions, that could help moving this paper forward. 

Response: We appreciate the reviewer effort for improving the manuscript. We have incorporated all comments and addressed each one in detail below. 

Typesetting and formatting:

I don’t normally pay too much attention to these aspects at this stage. However, I strongly recommend the paper by checked for type setting errors, and the written structure of some paragraphs is reviewed to help the reader. Details are offered below indicating page and line numbers. Yet, as a direct note to the authors, in the pdf version I had access to there was no line number after page 13 of the document (after Table 1), and at line 275 (Table 1), the page number starts again with 1. For this section then I will be referring my comments using these numbers while trying to point out the paragraph number and a few words in the sentence to facilitate the review process.

Response: Done. We added continuous line numbers after line 274, until the end of the manuscript.

Title:

I suggest the title to be more directly linked with the results (e.g. Elevation and latitude drives structure and tree species composition in Andean forests). As I will mention later, the manuscript includes in the discussion a long portion about the value of the plot network that I am not sure if is relevant. However, if the authors decide to keep it, maybe a change in the title should also reflect this (e.g. Elevation and latitude drives structure and tree species composition in Andean forests: results from a large-scale plot network).

Response: We have changed the title as suggested. 

Abstract:

This section needs to be modified a bit. In particular, the reader will benefit from having more specific results being mentioned in the abstract. For example, there is an important aspect of the study related to plot size and its effects on the results that is not mentioned here. 

Response: Agreed. We added a sentence to refer to plot size effect: Overall, plots ~ 0.5-ha or larger may be preferred for describing patterns at regional scales in order to avoid plot size effects (P4, L81-82). 

Page 4, Lines 68 - 69: “…Here, we assessed patterns of Andean forest tree structure and diversity along ~ 4000 km of latitude and ~ 4000 m of elevation range…”. Consider changing to: “…In this study, we assessed patterns of structure and tree species diversity across a wide altitudinal and latitudinal range in Andean forests…” 

Response: We changed it as suggested. 

Page 4, Line 75: “…Subtropical forests have distinct composition from tropical forests...” 

Consider changing to: “…Subtropical forests have distinct tree species composition compared to those in the tropical region…”

Response: We changed it as suggested.

Introduction:

Page 4, L83: “Indeed, the Tropical Andes…”. The region is already mentioned in the first sentence. Just start the line with something like “…Indeed, this region is one of the most diverse terrestrial hotspots on earth [1]…”

Response: We changed it as suggested.

Page 5, L86: “…that have not been described yet…” Change to: “…that have not yet been described…”.

Response: We changed it as suggested.

P5 L87-90: There are two different sentences, with different citations too, that mention “climate regulation”. Unify or combined these statements avoiding repetition. 

Response: We combined both sentences.

P5 L92-95: Again, two sentences discuss similar aspects, specifically about social aspects (i.e. “population growth” and “economic, and cultural factors”). Consider simplifying this paragraph. Also, is not clear what “contrasting patterns of human population growth” means? Please clarify. 

Response: We combined both sentences.

P5 L96: “Changes to Andean forests…” Changes in what? Forest cover? Structure? I don’t think the word “patterns” fits here. Consider something like: “…Changes in Andean forest cover includes both forest expansion, mostly frequent above 2000 masl) and deforestation that often dominates at lower elevations [12]…”

Response: We changed it as suggested.

P5 L102: “…are already undergoing…” Change to: “…experiencing an apparent shift in species composition (i.e…)” 

Response: We changed as suggested. 

P5 L104: Are these predicted shifts for plant species, animal species? Please clarify.

Response: Predicted shifts are for vascular plant species. We clarified it.

P6 L109-114: I am not sure if this whole paragraph fits here. Are those “knowledge gaps” related to the study? I understand the importance of collaborative research networks but the next paragraph (L116) immediately start discussing the effects of elevation on tree density. Consider removing this paragraph or in any case a connecting sentence is needed before you jump to the density portion.

Response: Knowledge gaps are related to the study. We clarified this, and rephrase the sentence adding connection to next paragraph (P6 L121-125). 

P6 L121 “…subtropical section of the Andes…” Just say” “subtropical Andes…” 

Response: We changed as suggested. 

P6 L128: “Species richness…” All species? Plants? 

Response: We referred to plant species. We clarified it.

Clarify. P7 L134: “…also showed…” -- > “…also shows the…” 

Response: We changed as suggested. 

P7 L135: “…being maximum…” -- > “…reaching a maximum at…”. 

Response: We changed as suggested. 

P7 L136: “…Argentina [38], decreasing…” -- > “…Argentina [38], while decreasing above…”. 

Response: We changed as suggested. 

P7 L138-140: I agree that long-term monitoring is needed, but the study do not use or report temporal trends so how is this relevant here? Consider moving to the conclusions. 

Response: We deleted as suggested.

P7 L142: “…the main structural and diversity patterns…” -- > “…the main patterns of structure and tree species diversity in Andean forest communities…”

Response: We changed as suggested. 

P7 L144: Why consider species richness as a structural component? 

Response: Actually, we consider species richness within diversity patterns. We clarified this. 

P7 L148-150: “…This study…” This paragraph would fit better perhaps at the beginning of this section in L142. Also, is not clear what do you mean about “…at regional scales in global context...”? Perhaps just say “…This study is one of the first attempts to describe and characterize regional patterns of forest structure and diversity in the Andes across (go to L143)…” 

Response: We moved the sentence to the beginning of the paragraph and changed as suggested. 

Two major points here: 1) I think is important to highlight how is this study different from the Baez et al study? I understand that in the 2015 paper there was no consideration about diversity and here you use a much higher number of plots. Thus, I see the need for making this point clear; 2) I think at the end of the introduction some general hypotheses would be useful. What were the patterns expected considered earlier evidences? 

Response: We added a sentence highlighting the progress of this analysis in relation to Baez et al 2015, as suggested (P8, L157-158). Also, we added a general hypothesis and two specific ones at the end of introduction section (P8, L166-172).

Materials and Methods:

P8 L155: “…Paleogene (). Subsequent…” Change to “…Paleogene (), and subsequent collisions…”. Response: We changed as suggested. 

P8 L156: The reference of Hoorn et al, 2010 does not follow PLoS guidelines (i.e. []). P8 L157: “…late middle…” -- > “…late mid Miocene…” 

Response: We changed as suggested. 

P8 L160: A citation is needed for this paragraph after “…Amazon ecosystems…”. 

We added citation. 

P8 L164: “…biodiversity and distribution…” -- > “…biodiversity and species distributions…” 

Response: We changed as suggested. 

P8 L169: “…vary gradually…” how? What does the word gradually means? I think the paragraph that starts later in L180 of page 9 with “...The mountain forest ecosystem…” should be moved here in support of this initial statement. In relation to this, I liked Figure 1, but I also thinks it deserves a better explanation, either in the main text or at least as an expanded figure caption. Why these variations in forest architecture? Some lines on temperature and radiation effects would be useful. 

Response: We reorganized the paragraph, including suggestions for a better understanding (P9 L191-206). Also the paragraph includes lines on temperature and radiation. 

P9 L194: Consider using “is” instead of “has been” when referring to the objective of the network. Response: We changed as suggested. 

P9 L196-197: Delete “the” before exchange, development, strengthening. 

Response: We deleted as suggested.

P10 L205-212. This whole section and the next (L215 to 212) about the protocols can be combined, simplified and shortened. 

Response: We simplified, combined and shorten both sections as suggested. 

P11 L226: “…events and recorded and the DBH of marked trees are…” -- > “…events are

recorded and the DBH of marked trees is remeasured…” 

Response: We changed as suggested.

P11 L230-241: I understand the importance of this section and I hugely support collaboration and networking. I am trying to find sections that could be simplified to allow for more space for discussion and I think this is definitely one of those parts. This paragraph here seems out of place. It could be shortened and added to the background section, used in a new appendix to fully describe the work of the Andean network or simply removed. As mentioned earlier when reviewing the title of the paper, if the goal is to highlight the value of the network a different approach would be useful, where a more in-depth discussion about research collaboration and networking is part of the manuscript. 

Response: We deleted this section as suggested by Reviewer 1.

P12 L247: Are these values in precipitation 1901 ± 903 refers to the mean and SD? Clarify. 

Response: We clarified as suggested. 

P12 L248: “…mature forest but the Network also includes a few older…” -- > “…mature forests but the network also includes several older… 

Response: We changed as suggested.

P12 L250-252: This statement about the data being available is out of place. This should be part of the data availability statement that is one of the specific sections asked by PLoS. Also, with regards to this there are three different statements: In Data availability the authors express: Yes - all data are fully available without restriction Later, when answering the questions on where this data could be found the authors expressed “All relevant data are within the manuscript and its Supporting Information files…”. Delete the portion in the main text and unify these statements clearly. 

Response: We deleted the incorrect statement in text as suggested and clarified that: “All relevant data upon which all the presented results in this manuscript are based on is included in its Supporting Information files…” (P13, L296-300). In this sense, as mentioned before in Table S1 we included: i) number of stems, ii) basal area (both extrapolated to 1-ha) and iii) species richness for the 491 plots. Additionally, in Appendix 1 we reported the abundance of tree species per country. As stated before, this information constitutes the primary input used in the analyses to describe the main structural and compositional trends across the Andean forests. We clarified this in text (P13, L296-300).

P12 L254: Again, the values in plot size refers to mean and SD? I don’t understand 0.32 ± 0.47 ha? How is this possible?. The same observation applies to the subsequent parts where plot size is mentioned. Also, I don’t think is necessary to put ± 0 whenever the plot size remains constant in some regions/countries. 

Response: We clarified that values in plot size refers to mean ± SD, and deleted ± 0.

P12 L261: “…being the longest subset…” -- > “…being the longest known subset…” Changed

P13 L270 or caption in Figure 2: Please clarify what do you mean when you say “…One (1) census refers to the establishment of a permanent plot”? I understand a one census plot as that with only one measurement, whether temporal or permanent.

Response: We deleted the phrase as it was confusing. Exactly, one census refer to one measurement, whether temporal or permanent. 

L275 Table 1 (No page number! We added page number): It is not clear to me that if the range shown for plot area also corresponds to the range in basal area. That is for example: Row 1 in the table (AR). A 0.16 ha plot accounts for 78 stems and 5 m2 in BA, while the 6-ha plot is linked to 1834 stems and 189.4 m2? Why not just simply express everything in 1 ha scale? I understand the issues of extrapolation, especially from really small plots, but yet you still perform the analysis later. Please clarify. 

Response: The range shown for plot area not necessary corresponds to the range in stems or basal area. In this table we prefer to show plot values, and not extrapolate to 1-ha in order to avoid the very large variability and dispersion when scaled to 1-ha (e.g. 0.01 ha plot with 1 stem would extrapolated to 100 stems ha-1, and in a 0.04-ha plot with 35 trees extrapolated value of basal area would be as high as 131.2 m2 ha-1). We discarded small plot for the analyses. We clarified this in text: “To analyze patterns in stem density, basal area and species richness, we considered those plots ≥ 0.1-ha (n = 236) where stems ≥10 cm DBH were measured. Due to high variability and dispersion when extrapolating basal area and stem density at a larger spatial scale we discarded plots of 0.01-ha, 0.04-ha, 0.05-ha and 0.08-ha (total n = 255)” (P16, L331-335). This was also stated in the discussion section (P24, L498-501). 

*** Starting here there is no line number, and page number is 2 for the section where “Data analyses” subtitle is included. I’m using these numbers and will add the “p” for paragraph.

Response: We added line and page numbers. 

P2 paragraph 1: “…resolution that represents the…” -- > “…resolution covering the average…” Response: We changed as suggested.

P2 p1: “…from 1 C for Ecuador…” -- > “from 1 C in Ecuador…” Same for the rest of the sentences where a similar text is used (i.e. change “for” to “in”). 

Response: We changed as suggested.

Data Analyses 

P2 p2: Delete “In order to…” Just simply say “To analyze…” Changed

“…(n = 236) and stems…” -- > “…(n = 236) where stems ≥ 10 cm DBH were measured. Due to high variability and dispersion when extrapolating basal area and stem density…we discarded plots of 0.01 to 0.08 ha in size (total n = 255)...”

Response: We changed as suggested.

. 

Related to this part of the analytical approach, I am wondering how different are the extrapolations when you compare a 0.08 ha plot and a 0.1 one? You have discarded quite a lot of plots using this cut-off (which I am ok with) but I asked myself if maybe just 0.05 ha would have been enough?

Response: In addition to the high variability and dispersion of data when extrapolating basal area and stem density of small plots at a larger spatial scale, we chose the 0.1 cut-off point as it is a standard minimum size for plots in forestry, thus for comparisons. 

Citation [55] should be placed after the (GLM) portion as this work refers to this modeling approach. Response: We changed as suggested. 

“Considering the nature of the response variables…” what is that nature? Statistical distribution? Did you perform a distribution test ahead of the GLMs? Please clarify. 

Response: We referred that: “We used a Quasi-Poisson distribution for stem density and species richness …, and a Gaussian distribution with log-link function for basal área”. This is clarified in text (P17, L343-3415. We deleted the phrase “Considering the nature of the response variables”…to avoid confusion.

P3 p2: Delete “In order to…” -- > “To analyze patterns of tree species composition and…, we used all data available from all plots, and from all species fully identified (2341) with stems ≥ 10 cm DBH. To describe species composition, we applied a Non-Metric Multidimensional Scaling approach…” 

Response: We changed it as suggested. 

[…[61] using latitudinal bands as random factor…” -- > “…as a random factor…” 

Response: We changed it as suggested. 

P4: Include citations for all R packages used.

Response: R package used are cited in text: “All analyses were performed in R 3.4.3 [63], using AER to test overdispersion in GLM, vegan for ordination analysis, and lme4 for LMM” (P18 L372-373).

Results:

P4 p2: “… Mean annual air temperature and rainfall were positively correlated (r = 0.46, p <

0.001) (Fig 3B)…” I trust the numbers but the correlation in the figure seems to be driven by a few really wet points. Can you clarify this? Adding a trend line would be useful too. Using green and red dots here makes difficult to tease apart the countries. I suggest using an alternate color palette and perhaps different symbols for each country. 

Response: We have checked the correlation and it is correct. There are some plots that have between 2000 and 3000 mm of rainfall which also have around 25 degrees of temperature, but also some other plots with 4000 and 5000 mm of rainfall that also have around 23-24 degrees. We did not include a trend line as this is a correlation and no cause-effect is expected. 

P5 p1: Delete “Considering plots…” since is already mentioned in the methods. Just start with

“we found that stem density…” 

Response: We changed as suggested.

With regards to this section and Figure 4, the small-sized plots seem to be creating a lot of noise in the trends. Why not testing stem density and basal area for only 1-ha plots as shown for species richness in Fig 4G? Also discussed later, are the plots with higher stem density also showing high basal area? There is a brief mention in the discussion about this relationship, but a simple bivariate plot would help. Response: We tested stem density and basal area for only 1-ha plots and found that results and tendencies were very similar. Thus we kept small plots in order to have higher number of plots. 

Plots with higher stem density not necessary have higher basal area. 

P6 p2: “…we registered a total of 97,054 tree stems’…” The abstract mentions 86,964 individuals (?). Also, are these all tree species? When authors say stems it might imply other life forms as palms or tree-ferns. Please clarify.

Response: The number 97,054 referred to all stems ≥10 cm while 86,964 referred to identified stems ≥ 10 cm diameter to species level. We clarified this in abstract (P4, L74-75). We addressed the entire manuscript to trees but actually we included palms and ferns. 

P6 p2: “…was shared among all data sets…” -- > “…was shared among all plots…” 

Response: We changed as suggested. 

P6 p3: “The study plots tended to segregate along…” Use a different word for segregate:

cluster, group. 

Response: We used the word “group” as suggested (P20 L426-438).

P6 p3: “…Elevation and latitude correlated with both…” -- > “…correlated well with…”. Also, in Figure 5 could you explain what does each axis in the NMDS represent? 

Response: We changed as suggested. 

Discussion and conclusions: See in bold edits and suggestions.

An overall recommendation here is that for every sentence/statement where the authors are highlighting a specific result, having a direct reference to the particular figure or table where the reader can refer to again would be quite useful. Also, I think authors can separate the conclusions here. 

P9 p1: “…it was less expected for rainfall as [5]…” Change the structure of the sentence and citation format to “…it was less expected for rainfall as Urrutia et al [5] reported for the Andes where rainfall did not covaried linearly with elevation…”. Also, this statement relates to an earlier observation about the need for some hypothesis statements. If you say “it was less expected”, what were your initial expectations? 

Response: Changed as suggested. Actually, we were referring that our findings were different from those of Urrutia et al. We rephrased the statement to avoid confusion (P23, L479-481). 

P9 p2: “Both stem density and basal area were related with the gradients addressed…” -- >

“Both stem density and basal area were related to the gradients covered by our dataset…”

Response: Changed as suggested. 

“…patterns of stem density were more consistent with elevation and latitude than for basal area…”

Response: Changed as suggested. 

Here, authors repeat some sentences from the results section. “Stem density peaked at 10-15 latitude”. 

Response: We rephrased the statement and differentiate it from the result section. 

“The observed increase in the mean values of the structural variables with elevation has

not…”

Response: Changed as suggested. 

P10 p1: “…or maybe some local climate and topographic conditions…” This sounds vague. What other climate or topographic conditions might have influenced these results? Explain. 

Response: We refer to some slopes orientation which may imply wetter or drier conditions, for example. We explained it in text as suggested (P24 L495-496). 

Land-use history is briefly mentioned as a potential driver of stem density. Yet, the authors have some (limited) information on the time since disturbance for some of the plots. Why not discuss this better? Can you filter some of the results based on different disturbance periods? Very dense plots might be a reflection of recent (or not that recent) events. At least in the form of an Appendix this would add some support when contrasting such a wide range of sites and conditions. 

Response: We do not have this information that is why we did not reported it. Nevertheless, we discussed that for the secondary plots considered, it may have had some influence on stem density (P24 L497-498). 

P10 p1: Change “border effect” for “edge effect”. Response: Changed as suggested.

Also, briefly explain why these effects are more pronounced for small plots?. Why plots ~0.5 ha or larger might be preferred? This section discussing potential effects of plot size also needs some citations. See for example: Wagner et al. 2010 Biotropica, Volume 42 (6): 664-671 and some references therein.

Response: We added the citation as suggested.

P10 p2: I think only mentioning the main hypotheses (e.g. Janzen & Connel) is not enough here. How these hypotheses are related to the results found? 

Response: We explained in text as suggested. 

P11 p2: “…plots associated with low diversity of tree species…”

Response: We changed as suggested. 

P11 p2: Change the reference format for Kattan et al. 2004 accordingly. P11 p3: Delete “As well…” 

Response: We deleted as suggested. 

P12 p1: “…are unknown…” change to -- > “ are not clear [36]”

Response: We changed as suggested. 

There is quite an abrupt change from this last paragraph to the next that discuss the importance of the Andean forest plot network. What are some of the potential implications of the findings? What does it mean that some forests in the Andes have higher density than others? More carbon? Less carbon? What about diversity? Are these forests well protected? I am not asking for a detailed analysis but just a brief consideration to potential links of the results to management or conservation aspects.

Response: We added a brief consideration as suggested (P26, L543-548).

P12 p2: The enumeration used in this long paragraph is confusing. There are two sections discussing different aspects, yet, some are redundant (e.g. #4 and #8 about models). Consider rewriting.

Response: We re-wrote as suggested. 

P12 p2: “…improving the number of hectares per country…” -- > “…increasing the area covered per country…”

Response: We changed as suggested.

---

## [Editor Report · Decision Letter 1]

26 Mar 2020

Elevation and latitude drives structure and tree species composition in Andean forests: results from a large-scale plot Network

PONE-D-19-30260R1

Dear Dr. Malizia,

We are pleased to inform you that your manuscript has been judged scientifically suitable for publication and will be formally accepted for publication once it complies with all outstanding technical requirements.

With kind regards,

RunGuo Zang

Academic Editor

PLOS ONE
---

## [Editor Report · Acceptance letter]

30 Mar 2020

PONE-D-19-30260R1 

Elevation and latitude drives structure and tree species composition in Andean forests: results from a large-scale plot Network 

Dear Dr. Malizia:

I am pleased to inform you that your manuscript has been deemed suitable for publication in PLOS ONE. Congratulations! Your manuscript is now with our production department. 

With kind regards,

on behalf of

Professor RunGuo Zang 

Academic Editor

PLOS ONE